# Expanding the Spectrum of *KDM5C* Neurodevelopmental Disorder: A Novel De Novo Stop Variant in a Young Woman and Emerging Genotype–Phenotype Correlations

**DOI:** 10.3390/genes13122266

**Published:** 2022-12-01

**Authors:** Carla Lintas, Irene Bottillo, Roberto Sacco, Alessia Azzarà, Ilaria Cassano, Maria Pia Ciccone, Paola Grammatico, Fiorella Gurrieri

**Affiliations:** 1Research Unit of Medical Genetics, Department of Medicine, Università Campus Bio-Medico di Roma, Via Alvaro del Portillo 21, 00128 Rome, Italy; 2Operative Research Unit of Medical Genetics, Fondazione Policlinico Universitario Campus Bio-Medico, Via Alvaro del Portillo 200, 00128 Rome, Italy; 3Division of Medical Genetics, Department of Experimental Medicine, San Camillo-Forlanini Hospital, Sapienza University, 00185 Rome, Italy

**Keywords:** X-linked intellectual disability, neuropsychiatric disorders, *KDM5C* gene, epigenetic signature, Claes–Jensen syndrome

## Abstract

As a consequence of the implementation of NGS technologies, the diagnostic yield of neurodevelopmental disorders has dramatically increased during the past two decades. Among neurodevelopmental genes, transcription-related genes and chromatin remodeling genes are the most represented category of disease-causing genes. Indeed, the term “chromatinopathies” is now widely used to describe epigenetic disorders caused by mutations in these genes. We hereby describe a twenty-seven-year-old female patient diagnosed with moderate intellectual disability comorbid with other neuropsychiatric and behavioral issues carrying a de novo heterozygous stop variant in the *KDM5C* gene (NM_004187.5: c. 3847G>T, p.Glu1283*), encoding a histone demethylase that specifically acts on the H3K4 lysines. The gene is located on the X chromosome and has been associated with Claes–Jensen-type intellectual disability, an X-linked syndromic disorder. We discuss our case in relation to previously reported affected females harboring pathogenic mutations in the *KDM5C* gene with the objective of delineating genotype–phenotype correlations and further defining a common recognizable phenotype. We also highlight the importance of reverse phenotyping in relation to whole-exome sequencing results.

## 1. Introduction

Claes–Jensen type (OMIM#300534), an X-linked syndromic neurodevelopmental disorder, is caused by mutations in the *KDM5C* gene (OMIM*314690). Maternally transmitted or, more rarely, de novo pathogenic variants explain about 0.7–2.8% of X-linked intellectual disability [1]. *KDM5C* encodes a histone demethylase that specifically acts on lysine 4 of histone H3 (H3K4), regulating transcriptional repression and chromatin remodeling. The gene is intolerant both to loss of function and missense mutations. Indeed, pathogenic nonsense, missense, and frameshift variants have been identified in several patients [2]. To date, about 60 pathogenic variants and 40 likely pathogenic variants are included in the ClinVar database (https://www.ncbi.nlm.nih.gov/clinvar/ (accessed on 1 November 2022). Moreover, recent studies have shown that patients with pathogenic variants display a specific peripheral blood epi-signature [3] that could represent a marker of pathogenic *KDM5C* variants [4].

The *KDM5C* knockout male mouse displays cognitive impairment, dendritic spine morphological abnormalities, and transcription dysregulation of many neurodevelopmental genes. The female knockout mouse exhibits a milder cognitive phenotype characterized by memory deficits and learning disability [5,6]. Similarly, heterozygous carrier females are often asymptomatic or mildly affected compared to hemizygous carrier males who usually exhibit moderate to severe intellectual disability and other neurodevelopmental disabilities including language or behavioral impairments and epilepsy [2]. Physical anomalies reported in the OMIM database (https://omim.org/ (accessed on 1 November 2022) include short stature, small forehead, prognatism, micrognathia, maxillary hypoplasia, facial hypotonia, flat philtrum, thin upper lip, high narrow palate, small and deep-set eyes, and large ears. Visual problems such as strabismus, hypermetropia, myopia, and skeletal anomalies of the hands and feet have also been reported in some affected individuals.

The clinical variability among females, even within the same family [2], might be related to the X chromosome inactivation pattern, which can help to compensate for disruptive genetic variants. However, a recent study [7] highlights a new role for KDM5C demethylase in activating the expression of Xist lncRNA (which is normally required for stable X chromosome inactivation), suggesting an alternative molecular disease mechanism based on a cascade effect of altered XIST expression on other X-linked genes. Therefore, in the presence of a KDM5C heterozygous variant, the expression of Xist could decrease, and this in turn could impair appropriate X chromosome inactivation. In this scenario, in female carriers, nonsense variants leading to the loss of function of one KDM5C allele could have a stronger and more detrimental effect on the X chromosome inactivation than missense variants. This in turn could result in a more severe phenotype in females bearing variants that abolish the expression of one allele, compared to females bearing missense variants. In order to explore this hypothesis, we reviewed genotype–phenotype correlations in all the affected females reported in the literature to date with a thorough clinical description. Moreover, we included in this dataset a twenty-seven-year-old female evaluated in the Medical Genetics outpatient clinics of our hospital. She carries a new de novo nonsense variant (NM_004187.5: c.3847 G>T, p.Glu1283*) in the *KDM5C* gene. We also aim at delineating specific clinical features that can help the geneticist to suspect or even recognize the *KDM5C*-related syndrome. Recognition of these features could drive reverse phenotyping analysis and facilitate the interpretation of pangenomic results.

## 2. Materials and Methods

### 2.1. Genetic Analyses

Whole exome sequencing (WES) was carried out on genomic DNA extracted from peripheral blood by the Nextera DNA Exome (Illumina, San Diego, CA, USA) on a NextSeq550Dx sequencer (Illumina). Sequencing reads were aligned to the human reference genome (UCSC hg19) by BWA (v0.7.7-isis-1.0.2) (Illumina). Variant calling was performed by GATK Variant Caller (v1.6-23-gf0210b3). DNA variants were annotated by eVai v2.5 (EnGenome). Variants mapping in genes associated to the following Human Phenotype Ontology (HPO) phenotypes were prioritized and then filtered by MAF < 0.01 (GnomAD v2.1): HP0001249, 0001256, 0002187, 0002342, 0006887, 0006889, 0010864, 0012759. Filtered variants were classified according to ACMG-AMP criteria [8]. The most phenotype-fitting variant was confirmed by Sanger sequencing. Genomic DNA from both parents was analyzed for segregation analysis of the variant.

### 2.2. Analysis of the X Chromosome Methylation Status

Considering the severity of clinical manifestations, an assay for evaluating the methylation status (i.e., inactivation) of the X chromosomes (X chromosome inactivation, XCI) was performed on DNA from blood leukocytes of the patient. The XCI pattern was determined by evaluating the cytosine methylation of CpG dinucleotides within the polymorphic CAG repeat in the first exon of the androgen receptor (*AR*) gene, located on the X chromosome [9]. In brief, DNA was digested by methylation-sensitive restriction enzymes (i.e., HpaII). After digestion, DNA amplification can occur only in the presence of methylated restriction sites (inactive allele). Both a digested and an undigested DNA sample were then amplified by two different primers’ pairs specific for the *AR* locus. PCR products were separated on an ABI3500 sequencer (Thermo Fisher Scientific, Waltham, MA, USA) and analyzed by GeneMapper 5 software (Thermo Fisher Scientific). All samples were analyzed in triplicate, and the average values were used in calculating the degree of X inactivation. In our heterozygous patient, the XCI skewing ratio was determined by comparing the ratio of allele peak heights in the digested sample (d1 and d2, for paternal and maternal alleles, respectively) with the ratio in the undigested sample (u1 and u2), according to the following proportion:

XCI percentage of the paternal allele = [(d1/u1)/{(d1/u1)+(d2/u2)}] × 100, which ranges from 0% to 100% [10].

### 2.3. Retrospective Study

We introduced the following key words in PubMed for *KDM5C* or *JARID1C* genes: (a) name of the gene; (b) name of the gene AND language delay; (c) name of the gene AND neurodevelopment; (d) name of the gene AND intellectual disability. We selected 12 papers [4,11,12,13,14,15,16,17,18,19,20,21] relative to patients carrying variants in *KDM5C*/*JARID1C* genes. Among them, we discarded papers without a deep clinical description and papers in which patients were only males. We ended up with four papers [4,17,19,20] concerning female patients carrying *KDM5C* mutations and with a thorough clinical description. We considered “thorough” to mean a clinical description where there was information concerning at least the intellectual status of the patient (developmental delay or ID), the physical description (peculiar physical anomalies), and information about language development and behavior. Most of the selected patients (Table 1) also had additional clinical information. We extrapolated genetic and clinical data in order to delineate possible genotype-phenotype correlations.

## 3. Results

### 3.1. Clinical Presentation of the Patient

The proband came to our genetic counselling service for a genetic re-evaluation when she was twenty-seven years old (Figure 1). She had previously undergone conventional karyotype and array-CGH, which were both normal. The initial “gestaltic” diagnosis was that of Rubinstein–Taybi syndrome when she was in her childhood, but no test was available at that time. She was the second child of two apparently healthy unrelated parents and was born naturally at term (Figure 2a). Perinatal distress due to umbilical cord wrapping and an Apgar index of 8, at the first and fifth minutes, was reported by the mother. Due to the low birthweight (2400 Kg) and due to respiratory and cardiopulmonary distress, she was hospitalized in intensive care; length was normal (48 cm). Initial language and motor development were referred to as normal: at six months, she sat down without support. Regression started at eight months, immediately after an epileptic seizure. Both cognitive and motor development were delayed: some communication skills were acquired at 24 months, and walking alone occurred at 18 months. EEG examination showed slight but not significant anomalies, whereas brain MRI was normal. Menarche was premature and occurred when she was 9 years old.

When she attended primary school, she received a diagnosis of mild intellectual disability and required special educational support for the entire school period. As a child, she was hyperactive with a short attention span, easy distractibility, and poor abilities in executive functions. Significant difficulties were also reported in expressive language, especially for lexicon and phonetics, as well as in calculation/arithmetic skills.

Clumsiness and deficit in fine motor skills were also reported. All these symptoms persisted throughout high school. As an adult, she also displays issues in the self-regulation of emotional response (outbursts), with agitation when frustrated, deficits in social skills, occasionally aggressive behavior, and indolence. Very low self-esteem and full awareness of her condition were also reported by the mother. She is no longer hyperactive but sometimes shows oppositional behaviors. She lives with her parents and carries out voluntary work in the local church. The last psychodiagnostic evaluation, using WAIS-R battery, rated an IQ of 47 and an overall evaluation of the global functioning equal to 45 (VGF). She has been constipated since childhood; she is also tending towards overweight and shows a strong preference for sweets.

Morphological evaluation showed a short stature (145 cm), a V-shaped frontal hairline, prognathism, facial hypotonia, a flat philtrum, a prominent nose with a downturned and bulbous tip, thin and arched eyebrows, small eyelashes, small deep-set eyes, strabismus (alternate exoforia), a thin upper lip, a high narrow palate, small feet, and squared hands with short and thick distal phalanges and brachydactyly (Figure 1). All these features have been described in patients with Claes–Jensen syndrome.

### 3.2. Genetic Findings

By WES, the proband was found to carry a nonsense variant in *KDM5C* (NM_004187.5:c.3847G>T; p.Glu1283*) in heterozygosity. The variant was classified as pathogenic since it is a null variant in a gene whose loss of function is a known mechanism of disease and is rare, being absent in from GnomAD 2.1 population database. By Sanger analysis, the presence of c.3847G>T was excluded in the DNA from peripheral blood of the parents, and thus the variant was assumed to be de novo. XCI analysis performed on DNA from proband’s lymphocytes showed a skewed pattern of methylation with 87% methylation status of one X chromosome allele (Figure 2b).

### 3.3. Genotype–Phenotype Correlations

The genetic and clinical characterization of twenty-three affected females carrying *KDM5C* variants is reported in Table 1. *KDM5C* pathogenic variants identified to date are summarized in Figure 3. There are thirteen females belonging to four families (family no. 4 with five females, family 10 with four females, and families 11 and 14 with two females each). Four females (17%) had no intellectual disability, whereas in the others (83%), mental impairment ranged from mild to moderate. No patients with severe intellectual disability were present in this cohort. These data confirm that females are less severely affected than males, as previously reported [2]. Language delay was reported in 18/23 females (78%), and in most cases, it was specific for expressive speech impairment (19). Overweight/obesity were common in older females (≥17 ys), affecting nine out of twelve patients (75%). In addition, specific behavioral problems, including aggressive behavior, low frustration tolerance, anxiety, and socialization impairment, seem to manifest mainly in adulthood (females ≥ 17 ys) and were reported in 7/12 patients (60%). Seven different types of missense variants (five in the JmjC domain, one in the JmjN domain, and one in the ARID domain) and seven types of frameshift (three), splicing (two), and nonsense (two) variants are reported in Table 1. Eleven females carry missense variants, and twelve females carry frameshift/truncating/splicing variants. In general, females carrying missense variants seem to be less severely affected than females carrying frameshift/truncating/splicing variants: normal development or mild intellectual disability was reported in 73% (8/11) females, whereas moderate intellectual disability was reported in 67% (8/12) females. In addition, if we consider only sporadic cases diagnosed by WES and not by segregation analysis as familial cases, the genotype–phenotype correlation becomes stronger: 83% (5/6, patients no. 1, 9, 10, 11, and 13) females diagnosed with moderate intellectual disability had stop/frameshift/splicing variants, and 100% (1/1, patient 2) of females diagnosed with mild intellectual disability or normal intelligence had missense variants (Table 1 and Table 2). Variable clinical expressivity was observed in all the four families reported in Table 1 and Table 2 and could be due to genetic background effects.

## 4. Discussion

Our study further contributes to outlining the specific phenotype in females with X-linked intellectual disability, Claes–Jensen type: variable degrees of developmental delay/intellectual disability (from absence to moderate), language delay, language impairment mainly in the expressive domain, and physical features (facial gestalt and short stature are the most frequent). Additionally, behavioral issues such as aggressive behavior, low frustration tolerance, anxiety, and social disability are often reported in young women, such as our patient. Our observations are in line with a recent caregiver report on the characteristics of patients diagnosed with *KDM5C* variants [1]: all females had developmental delay and language impairment, 60% had intellectual disability, 70% had short stature, 56% had aggressive behavior, and 44% had anxiety disorder. The clinical geneticist should be aware of these specific clinical features, especially when attempting the interpretation of more variants emerging from WES analysis [22]. Indeed, reverse phenotyping has become a very useful approach in clinical genetics following the routine implementation of pangenomic tests.

Our data support a possible genotype–phenotype correlation with a milder phenotype associated with missense variants and a more severe phenotype associated with variants leading to the loss of the expression of one *KDM5C* allele. Indeed, in vitro functional studies [23] have demonstrated that the frameshift variant V1075Yfs*2 and the N-terminally truncating variant p.M1_E165del (family 4, Table 1) completely abolish the *KDM5C* histone demethylase activity, whereas two missense variants (p.D402Y and p.P480L) reduce the activity. In agreement with our conclusions, Rujirabanjerd and colleagues (15) suggested a possible genotype–phenotype correlation in two unrelated families, each with three affected males. In the first family, males had a frameshift variant (p.K1087fs*43) and were all severely affected, whereas in the second family, males had a missense variant (p.P544T) and were mildly affected. In addition, males belonging to the first family had epilepsy, short stature, and hypereflexia/spasticity, which were absent in the males of the second family. Functional characterization of these two variants [15] supported the genotype–phenotype correlation observed in this study since the missense alterations affected both the tri- and didemethylase activity of KDM5C, whereas the frameshift variant resulted in a premature termination codon and a protein lacking the C terminal end and the second PHD zinc-finger domain.

A specific peripheral blood genome-wide DNA methylation signature has been detected in patients with Claes–Jensen syndrome [3]. Nine genomic regions encompassing 1769 CpGs were found to be differentially methylated in patients carrying pathogenic *KDM5C* variants when compared to healthy controls. Furthermore, female carriers showed less pronounced but distinctive changes in the same regions. Some of the carriers had missense variants, and others had truncating variants. Unfortunately, no methylation differences have been analyzed between the missense and truncating mutations’ carriers. Moreover, it is not clear whether the carriers were all completely healthy, as stated by the authors, or if some were affected. Indeed, the same females described in the study by Schenkel and colleagues [3] are reported in a previous study [14], and some of them were described as mildly affected (three of the four p.A77T carriers). In order to confirm the emerging genotype–phenotype correlation observed in this study, it would be very useful to perform genome-wide DNA methylation in affected females carrying different types of *KDM5C* variants.

Indeed, epigenome signatures might be used in the future as biomarkers for diagnosis, as well as in assisting the reclassification of variants of unknown significance, as has been recently done [4]. The skewed pattern of methylation we observed in our patient suggests a possible episignature of the X chromosome. This should be confirmed by genome-wide methylation studies involving more patients.

In summary, this study has shown a possible genotype–phenotype correlation in females carrying pathogenic *KDM5C* variants. This correlation should be confirmed in a larger cohort and could possibly include genome-wide DNA methylation analysis. Furthermore, it is advisable for clinicians to perform deep clinical characterizations of patients in order to facilitate the reverse phenotyping process.

## Figures and Tables

**Figure 1 genes-13-02266-f001:**
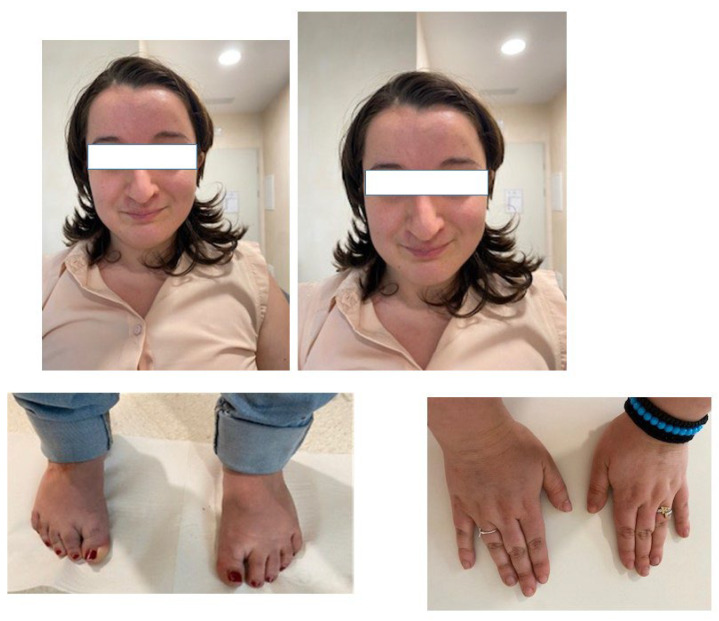
Clinical presentation of our patient.

**Figure 2 genes-13-02266-f002:**
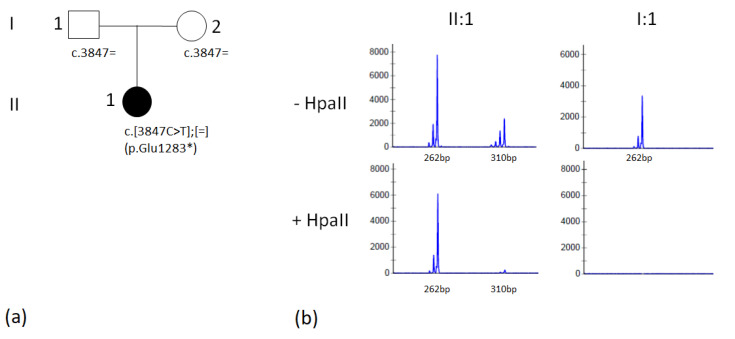
(**a**) Pedigree of the family; (**b**) analysis of X chromosome inactivation in our patient: fragment analysis results for the AR locus. Fragment analysis of undigested (− HpaII) and digested (+ HpaII) DNA from the proband (II:1) and her father (I:1). After HpaII digestion, DNA from the male individual (I:1) cannot be amplified with PCR, while DNA from the proband shows a skewed (about 87%) X chromosome inactivation.

**Figure 3 genes-13-02266-f003:**
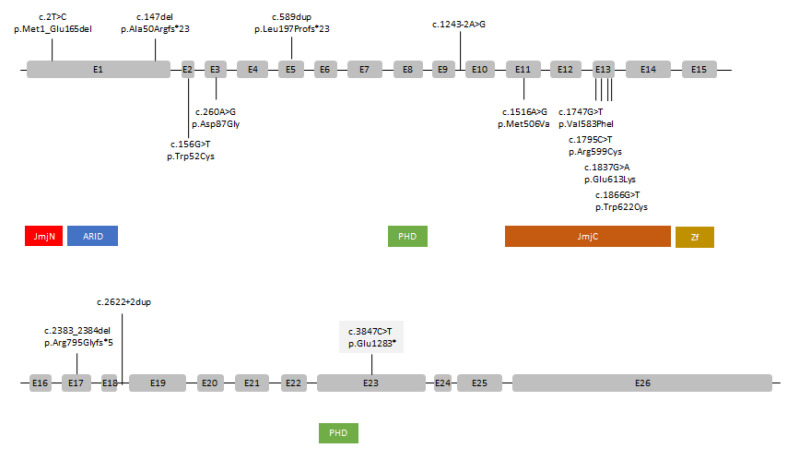
Schematic structure of the KDM5C gene and its known pathogenic variants. Exons are in scale; introns are not in scale. Exons belonging to isoform NM_004187.5 are shown by grey rectangles. The mutation found in the present patient is shown with a grey background. Truncating (i.e., nonsense and frameshift) and splicing variants are shown at the top, while non-synonymous and in-frame alterations are shown at the bottom. E: exon. Proteins’ functional domains are shown as colored rectangles under the gene scheme and include ARID: helix–turn–helix motif-based DNA-binding domain; JmjC: catalyzes demethylation of H3K4me3 to H3K4me1 JmjN: interacts with JmjC PHD: histone-methyl-lysine binding motif.

**Table 1 genes-13-02266-t001:** Female patients carrying pathogenic *KDM5C* variants with a detailed clinical history described to date. *KDM5C* reference: NM_004187.5.

Case	Fam No.	Familiar/De Novo	Mutation Type/Position	Detailed Clinical Features at Last Evaluation	Ref.
1	1	de novo	- Glu1283* c.3847 G>T - No in functional domain - end of protein	Evaluated at 27 years - moderate ID - short length (height 145 cm) - language delay, specific expressive language problems (lexicon and phonetics) still present - clumsiness and deficit in fine motor skills - widow’s peak at frontal hairline, prognathism, facial hypotonia, flat philtrum, prominent nose with downturned and bulbous tip, thin and arched eyebrows, small eyelashes, small deep-set eyes, strabismus (exoforia alternate), thin upper lip, high narrow palate, small feet, large hands with short and thick distal phalanges and brachydactyly - normal EEG and MRI - aggressive behavior, indolence, low frustration tolerance - constipation - overweight - skewed XCI (87%)	Our case
2	2	de novo	- Met506Val c.1516 A>G - in the jumonji-C domain - beginning of protein	Evaluated at 3 ys - no ID - normal length - language delay, specific expressive language problems (prosody) - ataxia and hypotonia - no dysmorphisms - normal EEG, MRI not done - no endocrine issues	[20]
3	3	inherited	- Val583Phe c.1747 G>T - in the jumonji-C domain - middle of the protein	Evaluated at 18 ys - no ID - short length (height 152 cm) - moderate speech impairment (vocabulary and reading speed), attention deficit - enophtalmia, horizontal palpebral fissures, cubitus valgus - overweight (62 kg, BMI = 27) - MRI normal	[4]
4	4	not known, mother	- Met1_Glu165del c.2T>C - beginning of protein	Evaluated at 42 ys - mild ID - short length (height 153 cm) - deep-set and almond-shaped eyes, divergent strabismus, a high and broad nasal bridge, high palate, mildly dysmorphic ears, brachydactyly, smiling face - obesity (BMI=32.5)	[17]
5	4	inherited, monozygous triplet	- Met1_Glu165del c.2T>C - beginning of protein	Evaluated at 16 ys - born prematurely (34 weeks) - moderate ID - short length (height 150.5 cm, −3 SD), weight 42.5 Kg (−2 SD), head circumference 53.5 cm (−1 SD) - high and narrow forehead, deep-set and almond-shaped eyes, epicanthic folds, strabismus, high and broad nasal bridge, high and narrow palate, mildly dysmorphic ears - joint laxity - slim fingers and toes, clinodactyly - skewed XCI (100%)	[17]
6	4	inherited, monozygous triplet	- Met1_Glu165del c.2T>C - beginning of protein	Evaluated at 16 ys - born prematurely (second) - no ID - short length (height 158 cm, −1.5 SD), weight 43 Kg (−2 SD), head circumference 53 cm (−1.5 SD) - high forehead, high and broad nasal bridge, high and narrow palate, mildly dysmorphic ears, slim fingers and toes - skewed XCI (100%)	[17]
7	4	inherited, monozygous triplet	- Met1_Glu165del c.2 T>C - beginning of protein	Evaluated at 16 ys - born prematurely (third) - mild ID - short length (height 155 cm, −2 SD), weight 43 Kg (−2 SD), head circumference 53 cm (−1.5 SD) - speech delay - high forehead, almond-shaped eyes, high and wide nasal bridge, high and narrow palate, mildly dysmorphic ears, slim fingers and toes - skewed XCI (100%)	[17]
8	4	inherited, daughter	- Met1_Glu165del c.2T>C - beginning of protein	Evaluated at 3.5 ys - developmental delay (speech delay and mixed type developmental problems) - short length (height 92.5 cm, −1.5 SD), weight 13.5 Kg (−1.5 SD), head circumference 47.5 cm (−2 SD) - deep-set eyes, a broad nasal bridge, a happily smiling face and mildly dysmorphic ears - MRI normal - skewed XCI (100%)	[17]
9	5	de novo	- Ala50Argfs*23 c.147del - beginning of protein	Evaluated at 5 ys - birth parameters: weight −2 SD, length −2.5 SD, OFC −0.5 SD - moderate developmental delay - severe language delay - high forehead, round flat midface, down-slanted palpebral fissures, broad tip of nose, short philtrum, maxillary hypoplasia, microstomia, small teeth, thin lips, short feet, short big toes, low columella, hypermetropia - normal MRI - low frustration tolerance - skewed XCI (95%)	[19]
10	6	de novo	- Leu197Profs*23 c.589dup - beginning of protein	Evaluated at 5 ys - birth parameters: weight normal, length −1 SD, OFC −1 SD - moderate developmental delay - severe language delay - abnormal EEG, one episode of absence - round face, low columella, thin lips, pes planus valgus, hypermetropia, astigmatism, amblyopia - constipation - low frustration tolerance, stereotypies, hyperphagia - skewed XCI (82%)	[19]
11	7	de novo	- c.1243-2A>G - splicing	Evaluated at 4 ys - birth parameters: weight −1.5 SD, length −1.9 SD, OFC −1 SD - moderate developmental delay - severe language delay - high forehead, large down-slanted palpebral fissures, marked sub-orbital crease, thin upper lips, 1 supernumerary tooth - tantrums, hetero-aggressive behavior, sleep disorder, some repetitive activities - constipation	[19]
12	8	de novo	- Trp622Cys c.1866G>T - in the jumonji-C domain - middle of the protein	Evaluated at 21 ys - birth parameters: weight −0.8 SD, length −2 SD, OFC −1.5 SD - moderate ID - severe language delay - orofacial dyspraxia - relational disorders, anxiety, insensitivity to pain - round face, short forehead, horizontal palpebral fissures, marked sub-orbital crease, broad nasal bridge, low columella, short philtrum, maxillary hypoplasia, thin lips, small irregular teeth, caries, short pes planus, genu valgum, hypermetropia - primary amenorrhea - diarrhea - severe obesity - skewed XCI (96%)	[19]
13	9	de novo	- Arg795Glyfs*5 c.2383_2384del - no functional domain - middle of the protein	Evaluated at 32 ys - birth parameters: weight normal, length −2 SD, OFC +1 SD - moderate ID - severe language delay - hetero-aggressive behavior, insensitivity to pain - marked sub-orbital crease, short philtrum, thin upper lip, prognathism, short square feet, brachydactyly of the toes - overweight - skewed XCI (91%)	[19]
14	10	inherited, daughter	- Asp87Gly c.260A>G - ARID functional domain - beginning of the protein	Evaluated at 45 ys - mild ID - anxiety - fine motor delay and memory difficulties - round face, broad nasal bridge, large columella, thin lips - glaucoma - overweight - random XCI (60%)	[19]
15	10	inherited, daughter	- Asp87Gly c.260 A>G - ARID functional domain - beginning of the protein	Evaluated at 45 ys - birth weight 2.750 Kg - moderate ID - language delay - anxiety - round face, round nasal tip, large columella, long philtrum - myopia - random XCI (50%)	[19]
16	10	inherited, niece	- Asp87Gly c.260 A>G - ARID functional domain - beginning of the protein	Evaluated at 17 ys - birth parameters: weight normal, length −1 SD, OFC normal - mild ID - language delay and fine motor delay and memory difficulties - low frustration tolerance, aggressive behavior, tantrums - round and flat face, high and prominent forehead, broad nasal bridge, short philtrum, thin lips - myopia - overweight - random XCI (70%)	[19]
17	10	inherited, niece	- Asp87Gly c.260 A>G - ARID functional domain - beginning of the protein	Evaluated at 6 ys - mild ID - language delay and fine motor delay - round face, broad nasal bridge - hypermetropia, convergence disorder - overweight - random XCI (60%)	[19]
18	11	not known, mother	- Trp52Cys c.156 G>T - JmjN domain - beginning of the protein	Evaluated at 55 ys - mild ID - language delay - normal behavior	[19]
19	11	inherited, daughter	- Trp52Cys c.156 G>T - JmjN domain - beginning of the protein	Evaluated at 24 ys - moderate ID - language delay - MRI: bilateral frontal hypersignals - normal behavior - round face, macrocephaly, marked sub-orbital crease, low columella, short philtrum, thin lips - atrial septal defect - skewed XCI	[19]
20	12	not known, mother	- Arg599Cys c.1795 C>T - JmjC domain - middle of the protein	Evaluated at 54 ys - no ID, learning difficulties - normal behavior - large face, broad nasal bridge, low columella	[19]
21	13	not known, daughter	- Glu613Lys c.1837 G>A - JmjC domain - middle of the protein	Evaluated at 36 ys - mild ID - language delay (dysarthria) - low frustration tolerance, aggressive behavior, compulsive overreacting - broad nasal bridge, low columella, short philtrum, maxillary hypoplasia, small and high arched palate - gynoid obesity - hypermetropia and left esotropia - random XCI	[19]
22	14	not known, mother	- c.2622 +2 dup	- mild ID - language delay - hypotonia - aggressive behavior - long face, large forehead	[19]
23	14	inherited, daughter	- c.2622 +2 dup	Evaluated at 7 ys - moderate ID - language delay - relational disorder, stereotypies - oculo-manual coordination difficulties, global hypotonia - round and flat face, high and prominent forehead, marked sub-orbital crease, broad tip of nose, microstomia, thin lips - random XCI	[19]

**Table 2 genes-13-02266-t002:** Correlation between mutation types and degree of intellectual disability in 23 females carrying pathogenic *KDM5C* variants.

Case	Fam No.	Familiar/ De Novo	Mutation Type	Degree of Intellectual Disability (ID)	Reference
1	1	de novo	-Glu1283* c.3847 G>T	MODERATE	Our case
2	2	de novo	-Met506Val c.1516 A>G	NO ID	[20]
3	3	inherited	-Val583Phe c.1747 G>T	NO ID	[4]
4–8	4	familiar	-Met1_Glu165del c.2T>C	Variable: MILD (in two patients), MODERATE, NO ID, DEVELOPMENTAL DELAY	[17]
9	5	de novo	-Ala50Argfs*23 c.147del	MODERATE	[19]
10	6	de novo	-Leu197Profs*23 c.589dup	MODERATE	[19]
11	7	de novo	-c.1243-2A>G -splicing	MODERATE	[19]
12	8	de novo	-Trp622Cys c.1866G>T	MODERATE	[19]
13	9	de novo	-Arg795Glyfs*5 c.2383_2384del	MODERATE	[19]
14–17	10	familiar	-Asp87Gly c.260A>G	Variable: MILD (3 patients), MODERATE	[19]
18–19	11	not known	-Trp52Cys c.156 G>T	Variable: MILD, MODERATE	[19]
20	12	not known	-Arg599Cys c.1795 C>T	NO ID	[19]
21	13	not known	-Glu613Lys c.1837 G>A	MILD	[19]
22–23	14	familiar	-c.2622 +2 dup	Variable: MILD, MODERATE	[19]

## Data Availability

Research data not shared.

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
