# Peer review of "Expanding the Spectrum of KDM5C Neurodevelopmental Disorder: A Novel De Novo Stop Variant in a Young Woman and Emerging Genotype–Phenotype Correlations"

_genes, 2022, doi:10.3390/genes13122266_

Round 1
Reviewer 1 Report
The authors have done an extensive clinical characterization of a female patient carrying a novel stop variant in KDM5C gene. The relationship established between the genotype and phenotype while comparing other contemporaneous studies with that mutation is commendable. I would like to see some of my following concerns addressed beforehand:
1) Inclusion of a pedigree chart would be highly essential for this article.
2) A figure showing the mutations in the coding region of the gene would be helpful. Please include all mutation types.
3) While the detailed description of each patient is shown in Table 1, it would also be beneficial to show a summarized table with breakdown of clinical characteristics grouped by severity of intellectual disability (mild, moderate and none). Statistical tests if possible would be a welcome addition.
4) Results from the methylation status should be either shown in a figure or results of the data should be attached in the supplementary material. Having both will be much desirable.
Author Response
Dear reviewer,
thank you very much for appreciating our work. I have enclosed a file with the answers to your requests.
Kind regards,
Carla Lintas

Reviewer 2 Report
This article reports a new patient with X-linked intellectual disability who carries a de novo nonsense mutation in KDM5C. This article also summarizes genotype-phenotype correlations from previous reports. This paper will be a good resource for researchers in this field. A few points should be addressed.
1. The authors should consider showing a figure of the XCI analysis.
2. Line 182-183, the authors should discuss in more detail what the 87% methylation status mean, i.e. less X chromosome inactivation and how does that lead to the disease. More importantly, the authors should discuss the correlation between between XCI and the phenotype based on table 1.
3. Line 200-203, the authors should rephrase the sentence to make it more clear that there are 11 females with missense mutations and 12 with frameshift/truncating/splicing mutations.
4. Line 206, the authors should point out which 5 females they are referring to in the table. From what I can see, patient #12 has moderate ID and a de novo mutation which is missense, which means that not 100% females diagnosed with moderate intellectual disability had stop/frameshift/ frameshift variants. Also what is the rationale for only looking at de novo mutations in this type of analysis?
5. Line 76-77, grammar error in the sentence "the expression of Xist could decrease and accordingly to the efficiency 77 of X-chromosome inactivation."
Author Response

(The authors gave the same response as above.)

Reviewer 3 Report
The manuscript was about the manifestations of a non-sense mutation in KDM5C gene in a female patient. I have some concerns and advices in this regard, which I hope can help the authors to improve their study.
1) The manifestations of the mutation were extensive and this was more similar to a syndrome and a chromosomal anomaly instead of a single mutation in a gene. Specially, it seems weird when we observe this in a heterozygote X-linked gene in a female. Even the authors indicated that the type of mutation was loss of function, while the patient still had a normal allele. Thus, it is more similar to a haplo-insufficiency mutation rather than loss of function. However, mosaicism may be the reason in such cases (and obviously this need to be proved by evidences) and the author referenced some studies in accordance of their results. I still believe that these are important questions that must be justified.
2) The presentation of results was very poor, ambiguous. especially in "Genotype-phenotype correlations" section. The authors did not indicate the genders of the patients and some other necessary details in other studies that they reviewed their results. The authors indicated a cohort, while reviewing other studies do not make their study into a cohort. This was a case report study supported by reviewing other similar studies. The results are obviously presented in a hurry and important details are missed, Specially when comparing their results with the results of other studies. This is important in understanding of concordance of the studies. For example, were the only mutations in patients of the reviewed studies those ones on KDM5C gene?
3) Informed consent from the patient herself is necessary, if she is not completely intellectually disabled and in custody of her family. The authors indicated that she had some high school educations.
4) English language of the study needs to be improved.
Author Response
Dear Reviewer,
Thank you very much for reading our manuscript.
I have enclosed the answers to your comments.
Kind regards
Carla Lintas

Reviewer 4 Report
The authors showed a specific case with KDM5C mutation, which provided evidence for phenotype-genotype correlation along with other published papers.
The difference between null or missense mutation is interesting and may play an important role in regulating demethylation. However, no more evidence (e.g., methylation differentiation analysis) to support that point, which the authors have already stated as a drawback. Also, the results mostly depend on the XCI percentage, which may not be a good indicator for X-linked variants. Please see below:
https://www.nature.com/articles/s41431-018-0291-3
This paper also pointed out that AR should not be used as the only gene for the indicator of methylation, and several more genes should also be included.
Moreover, for the monozygotic triplet with the same mutation, the clinical manifestations (especially the IDs) are clearly different, suggesting other factors may play more roles, thus more discussion would be better.
The authors could introduce more about the XCI methodology and at least discuss the usage of a single gene as an XCI indicator or maybe their opinions on the included paper and other potentially interesting questions if no more experiments can be performed.
Author Response

(The authors gave the same response as above.)

Round 2
Reviewer 3 Report
The work could be improved by adding complementary experiments and adding similar cases. However, I understand limitations in this regard and the study still contains valuable medical information to help clinicians.
Reviewer 4 Report
This reviewer has no more suggestions.